# Climate Change Education: Mapping the Nature of Climate Change, the Content Knowledge and Examination of Enactment in Upper Secondary Victorian Curriculum

**Efrat Eilam \*** , **Veerendra Prasad and Helen Widdop Quinton** 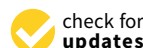

The College of Arts & Education, Victoria University, Melbourne 8001, Australia;
veerendra.prasad@live.vu.edu.au (V.P.); Helen.Widdop-Quinton@vu.edu.au (H.W.Q.)
\* Correspondence: efrat.eilam@vu.edu.au

**Abstract:** Climate change (CC) is widely accepted as the major threat of our time, posing unprecedented challenges to humanity. Yet very little is known regarding the ways in which upper-secondary curricula address the need to educate about this crisis. This study contributes to the field of CC education theoretically and empirically. From the theoretical perspective, the study contributes two CC conceptualisation frameworks: a characterisation of the *nature of CC*, and a mapping of the scope of CC content knowledge. The empirical contribution consists of examining CC education implementation within upper-secondary curriculum in the state of Victoria, Australia. Specifically we examined the CC conceptualisation and the scope of content present in the Victorian Certificate of Education (VCE) study designs. A total of 10 out of 94 study designs qualified for examination through referencing CC. The findings suggest that none of the study designs present a complete conceptualisation of the *nature of CC*. Common conceptualisations within the study designs perceive CC as a cause or an outcome, a problem of management, or of technological efficiency. CC content within the study designs is limited, and presents misconceptions, including the assumption that CC is a natural change caused by astronomical and solar systems. A cross-curriculum integration approach within the study designs is found to be ineffective. We conclude that CC presents a paradigm shift which brings about the new discipline of CC. There is a need for curricula reforms to address and incorporate CC as a coherent body of knowledge.

**Keywords:** climate change education; secondary curriculum; curriculum analysis; nature of climate change

## 1. Introduction

This paper theorises the conceptualization, the scope of content knowledge and the integration of climate change (CC) education. It continues to examine these aspects within the Victorian Certificate of Education (VCE) in the state of Victoria, as a case study of the Australian state and territory rarely researched upper-secondary CC curriculum [1].

Climate change is widely accepted as the major threat of our time, posing unprecedented challenges to humanity [2,3]. Yet very little is known regarding the ways in which curricula conceptualise CC, selected CC contents, and approaches to preparing graduates for dealing with the climate change crisis. The OECD [4] has expressed concerns that curriculum reforms suffer from time lags between the recognition that a change is needed to the actual implementation in curricula. For some major paradigm shifts and scientific discoveries the time lag between discovery and implementation in school curricula may span over many generations.

For example, Einstein published his General Theory of Relativity in 1915. His equations 'did away with Newton's theory of gravity and replaced it with curved space and warped time' [5]. Yet, 104 years later, this fundamental paradigm shift is still not fully reflected in primary and middle-school curricula in Australia, as well as in other countries. Foppoli, et al. [6] criticize Australian early school physics curricula for continuing to be dominated by Newtonian physics – with its absolute time, fixed space and lack of gravitational waves. This situation leaves the majority of graduates who do not continue to study physics into the final years, Years 11-12, with a knowledge of physics discoveries that is no more recent than 1865 [7].

Similarly, Charles Darwin published his seminal book, the 'Origin of species by means of natural selection' in 1859. The book became an immediate sensation and was widely accepted by the scientific community [8]. Yet it was only 122 years later, in 1981 in the United States of America, that the National Center for Science Education was founded to advocate the teaching of evolution in public schools. Now in the twenty first century, regardless of the theory's wide acceptance, its implementation in school curricula is still being contested in the American courts [8].

Many reasons may be attributed for the time lag. Foppoli, et al. [6] named a few. These include: the need to develop appropriate programs and materials; the need to train teachers and obtain their support; and, the need to gain public support. Other reasons may be attributed to deep rooted belief systems, as in the case of the evolution *versus* creationism debate [9].

When it comes to climate change science, similar to other important paradigm shifts, the science underlying this realization accumulated incrementally over centuries. Perhaps the first piece of scientific discovery related to this topic, may be attributed to the French physicist, Joseph Fourier, who discovered the Earth's natural greenhouse effect in 1824 [10,11]. In 1938, using records from 147 weather stations around the world, the British engineer Guy Callendar showed that temperatures had risen over the previous century, and that this was accompanied by increases in carbon dioxide concentrations. Callendar's proposition that carbon dioxide accumulation causes warming, was widely dismissed [11].

By 1961, when Charles David Keeling produced data showing that the level of atmospheric carbon dioxide is steadily rising, the scientific community was ready for a paradigm shift [2,11]. It took another 27 years for the United Nations to establish the Intergovernmental Panel on Climate Change (IPCC) in 1988 [2]. Since then, the IPCC has been assessing the science related to climate change, and providing policymakers with regular assessment reports related to climate change implications, potential risks, and adaptation and mitigation options [12].

Since its establishment, the IPCC reports have become increasingly alarming. The fourth report published in 2014 unequivocally claims that 'human influence on the climate system is clear, and recent anthropogenic emissions of greenhouse gases (GHG) are the highest in history' [13] (p. 2). More so, 'continued emission of greenhouse gases will cause further warming and long-lasting changes in all components of the climate system, increasing the likelihood of severe, pervasive and irreversible impacts for people and ecosystems' [13] (p. 8). The IPCC asserts that these impacts are not going away, but rather are here to stay for centuries, even if GHG emissions are stopped today [13] (p. 14). We may have expected that this conclusive, evidence-based portrayal of the state of our world should form sufficient grounds for curriculum reforms advancing integration of CC education.

Indeed, in line with this fundamental paradigm shift, various international and regional organizations have stressed the importance of introducing CC education. For example, the OECD in its report entitled 'Trends shaping education 2019' has highlighted the importance of CC education, and stressed its dual role in both equipping students with the skills required to succeed in a globally changed world, and as a means for combating CC [14] (p. 1).

However, regardless of the fact that over the past half a century, CC has become broadly accepted as the most defining challenge of our time, there is scarcity in research regarding the extent of CC education implementation. In addition, little is known regarding the extent to which the factors

identified by Foppoli, et al. [6] which influence curricula implementation lag, are operative in regard to CC education.

The present article seeks to contribute to our understanding regarding CC education implementation by focusing attention on the Australian curriculum. We are interested in developing our understanding regarding potential gaps of various types. These include: gaps between understandings about the *nature of climate change* and representation of this within the curriculum; gaps between the scope of CC as it is understood by the scientific community, and the scope of CC present in the curriculum; and, the gap between what may be regarded as 'best practice' curriculum integration and the current approach to integration.

To address these gaps, the study put forward twofold aims: to develop the theoretical underpinning of the research, and to apply this theoretical framework to examine the Victorian curriculum Years 11-12 study designs. In regard to the theoretical aspect of the study, we ask:

1.  How can climate change be conceptualized as a body of knowledge, consisting of inherent characteristics that may be regarded as constituting the *nature of climate change*?
2.  What is the scope of climate change content knowledge, and how can it be mapped for curriculum development purposes?

In regard to the empirical aspect of the study, we ask: Within the Victorian upper-secondary curriculum—

1.  How is climate change conceptualized?
2.  What is the scope of CC content knowledge present in study designs identified as addressing CC?
3.  How is the cross-curriculum integration approach addressed by the examined study designs?

In what follows, we set the scene for our investigation by reviewing CC education perceptions by intergovernmental and governmental organizations, and conceptualisation and implementation within school curricula.

## 2. Literature Review

### 2.1. Climate Change Education from the Perspectives of International Organizations

Climate change education is perceived in many international treaties and declarations, as a largely untapped strategic resource for building resilient and sustainable societies [15]. Since the 1992 United Nations Framework Convention on Climate Change (UNFCC), the critical role of education in CC responses has been repeatedly stressed and promoted in most subsequent conferences of the parties (COP) [16]. The Lima Ministerial Declaration on Education and Awareness-Raising, adopted at COP 20, calls for including CC education in school curricula and development plans [17]. Article 12 of the 2015 Paris Agreement, reaffirms that: 'Parties shall cooperate in taking measures... to enhance climate change education, training, public awareness, public participation and public access to information' [18].

United Nations Educational, Scientific and Cultural Organization (UNESCO) plays a leading role in the active promotion of CC education. UNESCO and UNFCC summarize the role of CC education and the requisites for implementation as follows:

Education seeks to achieve profound, long-term changes in understanding, particularly among young people. It involves developing educational curricula, training of trainers and teachers and adequate pedagogies. The results of a successful programme would ultimately be a population whose deep-seated appreciation of the climate challenge leads to greater national action and commitment [19] (p. 14).

Regardless of these high expectations, studies and reports regarding curriculum development at national levels are limited and the existing evidence suggest only sporadic uptake of these ideas, as outlined in what follows.

## 2.2. Climate Change Education in Nations' Curricula

Studies related to CC education policy research are scarce [20]. Aikens et al., in their systematic literature review of policy research in the area of environmental and sustainability education, have noted the 'dearth of research that examined education policy in relation to climate change' [21] (p. 334). When such systematic studies exist, rarely do they extend to analyse CC education in curriculum documents [22]. Our search of the literature published in English and available in open sources affirms this claim. While we were unable to find any systematic analyses of national CC curricula, the literature does provide some evidence regarding CC education examples, as well as some illumination regarding governmental approaches to CC education curriculum development.

Schreiner et al. [23] differentiate between national curricula in which CC is explicitly addressed, such as in Norway and Scotland, and national curricula in which CC is implicit and taught only in one particular grade, such as in Sweden, Denmark, England and Wales. In England for example, it was reported that students begin to learn about CC when they are 14 years old [24]. In France, Arnould [24] reported that education for sustainable development (ESD) is taught at all levels across all subjects in state schools, although climate change is not taught as a subject until secondary level. However, when CC is taught at French secondary schools, a cross-curriculum approach is applied, in which CC is integrated into the syllabi of various subjects. Thus, aspects of CC are taught in geography, the life sciences, earth science, economics and technology, while ethical aspects of CC may be discussed in subjects such as philosophy or history. Arnould critiques this approach by stating that the idea of bringing the various disciplines to work together on the shared topic has not been realized [24] (p. 339).

In Finland, the new National Curriculum, that took effect in 2016, was developed with a view of bringing disciplines together to avoid fragmentation. It introduced phenomena-based learning organised around interdisciplinary, holistic selected phenomena. The foundational value framing the educational purpose, is an eco-social approach [25]. While this framework could potentially support CC education, it is yet unclear to what extent the phenomena-based approach provides a home for CC education. Lehtonen et al. [25] cautions that for some teachers, multidisciplinary CC education may be challenging to teach. Hermans [26] mentions that in both the old and the new curriculum, geography carries most of the responsibility for CC education in years 7–9.

Israel, since 2004, has mandated sustainability education at all educational levels, from preschool to teacher colleges [27]. Climate change is integrated in years 10–12 subjects within geography and environmental development; environmental sciences; biology; chemistry; and earth sciences [28]. The 2004 Director General's Directive, issued by the Ministry of Education and Sports and the Ministry of Environmental Protection, entitled 'Implementing Education for Sustainable Development in Israel's Education System' provided a basis for sustainability education developments within these subjects. A later 2010 directive added a new integrated program consisting of seven topics to be taught at all school levels. These include: Education for sustainable development (ESD) and sustainable living; pollution reduction and environmental hazards; climate change, air pollution and greenhouses gases (GHG); waste problem and solutions and sustainable consumption; alternative energy sources; water management in Israel and water crisis; and biological diversity and open spaces. This integrated sustainability program that includes climate change is offered to education systems from preschool to upper-secondary, however, participation is voluntary. The program is supported by an allocation of 30 hours for teachers' professional development [27].

In the U.S.A. the Framework for K–12 Science Education (NRC, 2012) emphasizes climate change education as a central aspect of science education for middle and high-school students [29]. Arnould reports that the guidelines are not mandatory and that they are somewhat vague, encouraging teachers to discuss climate change in the classroom [24] (P. 338).

In Australia, in 2009, the government published a National Action Plan entitled 'Living Sustainably: Government's National Action Plan for Education for Sustainability'. This Action plan has no statutory power and its application is not mandated. The plan does not specifically address CC education, but rather incorporates it within the broader topic of sustainability [30]. In the Australian

Curriculum for Foundation to Year 10, the topic of sustainability appears as a cross-curriculum priority. However, the term climate change is conspicuous by its absence throughout most of the Australian Curriculum, including within the fundamental sections of Key Concepts and Organizing Ideas and from most learning areas (subjects). Climate change is referred to only twice throughout the curriculum documentation as exemplars of sustainability issues to consider within the learning areas of technologies and languages. In technologies CC appears within the following context: 'The curriculum focuses on the knowledge, understanding and skills necessary to design for effective sustainability action taking into account issues such as resource depletion and climate change' [31]. In languages CC appears as follows:

> In this way, students develop knowledge and understanding about sustainability within particular cultural contexts. This is crucial in the context of national and international concerns about, for example, climate change, food shortages and alternative ways of caring for land and agriculture [31].

The absence of CC from the Australian Curriculum in Years 7–10 and its subsuming under the cross-disciplinary topic of sustainability has been raised as a concern by several authors [32–34].

Overall, this scattered evidence suggests that in the lower and middle years, when CC is addressed, there is a tendency to integrate it within sustainability education curricula and this in turn can be voluntarily introduced as a cross-curriculum topic, subject to teachers' discretion. Within secondary schools, CC education when specified, is to be integrated through a cross-curriculum sustainability approach, in which various aspects of CC, are taught separately in discrete subjects. Generally, in reports on nations' curricula, it is not clear what aspects of the complex topic of CC are taught in which subjects and what efforts are made to integrate CC education between the different subjects. More so, there is a lack of clarity as to what aspects are being replicated in the different subjects and what aspects are being neglected and left as 'holes' that obstruct students' ability to develop comprehensive and integrative understanding of the multi-system complexity of CC. It is a long-held tradition for curriculum documents to base themselves around core disciplines. These subject divisions, while seemingly effective and justifiable in relation to bodies of knowledge such as mathematics and history, become highly challenging in relation to CC education [23]. When considering the various curriculum subjects, it seems that not one of the traditional disciplines is able to house CC education comprehensively [35,36].

### 2.3. Climate Change Conceptualization: The Nature of Climate Change

Climate change can no longer be regarded as an eclectic phenomenon, spread across multiple disciplines. Over the past century, CC has gradually evolved as a body of knowledge. Similar to the emergence of the field of ecology in the 1950's, with its early beginnings spread across many disciplines, climate change too has evolved within the various subjects to form a discipline in its own right. In a 1977 paper, Eugene Odum [37], one of the founders of ecology quotes Novikoff (1945), who stated that 'equally essential for the purposes of scientific analysis are both the isolation of parts of a whole and their integration into the structure of the whole' [38] and by thus, metaphorically, providing a pathway for understanding the forest as more than a collection of its trees (p. 1289). Odum ends his paper with a call to move beyond ecology into broader merges, which include political sciences, social sciences and other disciplines, as a way of foraging into 'unresearched levels of thinking and action' (p. 1292). Today, 42 years since Odum's publication, the emergent field of climate change is doing just that. CC's emergence has caused a paradigm shift in our understanding of the essential inter-relationships between the economy, society, global politics and the natural environment, with implications for sustaining life on earth [39,40]. The authentication of climate change as a body of knowledge has by now become evident by the establishment of countless institutes, university departments and organisations to study and respond to climate change.

Bodies of knowledge may be characterised by their nature, referring to the ways in which knowledge is produced, applied, valued and evaluated, as well as other characteristics related to governing epistemological principles. In science education, the notion of the *nature of science* has been thoroughly examined and researched in the context of science curricula since the 1950's [41,42]. However, when examining CC implementation within the curriculum, the question of the *nature of CC*, to the best of our knowledge, has thus far not been addressed. Most studies addressing CC in the context of education, focus mainly on aspects related to instructional models, content knowledge and pedagogy (e.g., [43,44]), rarely conceptualising CC as a body of knowledge, with its typical characteristics and governing principles. We propose that theoretical discussions regarding the *nature of CC* are urgently needed for supporting further CC curricula development and analysis. Such conceptualisation is vital to counteract the current fragmentation of CC education and to enable educational policy and practice to move forward towards more successful CC curriculum implementation.

### 2.4. Models and frameworks for Climate Change Education

Models and frameworks for climate change education vary in their approaches. Some models emphasise the importance of the scientific knowledge basis. Others emphasise the integration between scientific and humanistic dimensions of CC. Yet further CC education models are based on values propositions for forming worldviews and supporting personal and social growth.

At the scientific-basis end of the spectrum, Shepardson et al. [45] developed a climate system framework for teaching about CC. The framework consists of seven interconnected principles, addressing the science of CC. These include: The sun is the primary source of Earth's energy for Earth's climate system; climate is regulated by complex interactions among components of the Earth system; life on Earth depends on, is shaped by and affects climate; climate varies over space and time through both natural and human-made processes; our understanding of the climate system is improved through observations, theoretical studies and modeling; superimposed over natural variability, human activities are impacting the climate system; and climate change will have consequences for the Earth system and human lives (p. 329). Shepardson et al.'s framework identifies nine key elements that need to be addressed when teaching CC. These include: What is a climate system? Climate and weather; the Earth and Earth's energy budget; system feedbacks; the Sun (solar radiation); atmosphere (troposphere); ice and snow; oceans; and land and vegetation (p. 335).

At the more integrative end of the spectrum of approaches, some models inter-relate and integrate between the science and the humanistic aspects of CC. Kagawa and Selby [39] proposed an agenda for climate change education consisting of the following key themes: Education (i) must address the root causes; (ii) apply interdisciplinary and multidisciplinary approaches; (iii) integrate global climate justice education; (iv) responses need to be local and global; and (v) needs to be a social and holistic learning process. In addition, they call for educators to urgently and radically think through the implications of the invisibility and uncertainty of CC.

Anderson [46] identified specific categories of content knowledge that must be addressed. These essentials include: Understanding of scientific concepts; knowledge of the history and causes of CC; knowledge of and ability to distinguish between certainties, uncertainties, risks and consequences of environmental degradation; knowledge of mitigation and adaptation practices; understanding of different interests that shape different responses; and ability to critically judge the validity of these interests in relation to the public good [46] (p. 194).

UNESCO [47] has applied an integrative CC education approach in their development of a teacher training resource, entitled: Climate Change in the Classroom: UNESCO Course for Secondary Teachers on Climate Change Education for Sustainable Development. The syllabus conceptualises CC as primarily a human existential problem, emphasizes the interconnectedness between the scientific aspects of CC and all other human-related aspects. The scope of this CC education model is described as encompassing three interconnected dimensions: Mitigation, adaptation and understanding of and attentiveness. Mitigation refers to 'identifying the causes of climate change and developing

the knowledge, skills and dispositions required for individual and societal change to rectify those causes' [47] (p. 5). At a basic level of understanding, the causes may be attributed to GHG. At a deeper level, understanding the causes require questioning of the economic-social-cultural and other human-related systems. Adaptation refers to 'building resilience and reducing vulnerability in the face of climate change impacts that are already happening' [47] (p. 54). Adaptation is considered as both, basic technical level learning (such as learning about draught resistance crops) and deeper level investigations (such as examining the food production industry). Finally, understanding of and attentiveness refers to the ongoing reinforcement of cognisance to the realities of climate change, the understanding of root causes and the invisible crippling nature of climate change [47] (p. 5). The three dimensions of the UNESCO model are complementary and perceived as underpinning self-transformation through the interactive processes of reflection and active engagement.

Tolppanen et al.'s [48] comprehensive CC educational model also aims to capture holistically the humanistic and scientific aspects of CC. However, this model situates knowledge about CC within broad educational aspirations related to shaping the learners' personal growth, including aspects related to students' identity formation, ethical choices, actions and other cognitive-emotional aspects. The structure of the model aims to knit together in a visual way eight aspects corresponding metaphorically with bicycle parts (p. 459); these include: (i) Back wheels: Thinking skills; (ii) front wheel: Knowledge; (iii) frame: Identity, values and worldview; (iv) chains and pedals: Action to curb climate change; (v) saddle: Motivation and participation; (vi) brakes: Operational barriers; and (vii) lamp: Hope and other emotions; (viii) and handlebar: Future orientation.

In close relation to the above model, Lehtonen et al. [49] summarise eight elements perceived as forming the basis of CC education. The elements present a set of epistemological, ontological and axiological views. For example, an epistemological claim states that 'our understanding and response to climate change is socially constructed' (p. 366) and 'reflection on embodied experiences and emotions, intuitive knowing are useful resources for rational thinking and learning' (p. 367). Ontologically, it is claimed that 'nature and culture are one entity' (p. 367). Axiologically, the elements assert that 'hope, courage and trust are strengthened through embodied, shared experiences' (p. 367). In addition to the philosophical perspectives, the model includes specific pedagogical recommendations such as applying arts-based learning as it 'unleashes creative potential and naturally combines different ways of knowing: Pre-conscious, intuitive and rational' (p. 367); and dialogical learning, as these 'situations offer open encounters where adults and young people learn from each other and together construct pathways for a sustainable future' (p. 367).

The study reported in this paper is based in the epistemological view that CC content knowledge consists of both humanistic and scientific knowledge. However, we depart from some of the CC education models suggested above, by differentiating between CC content knowledge and the educational process. Such differentiation is required by the aim of our study, which is to analyse curriculum documents. The Victorian Curriculum documents primarily focus on what to teach. Therefore, our analysis focuses on CC as a body of knowledge. We now turn to consider models for integration of CC Education within the curriculum.

### 2.5. Integrating CC Education within School Curricula

The question of how to integrate CC within the curriculum is closely related to its conceptualisation among curricula developers. Traditionally, curricula are organised around disciplines [49]. When it comes to CC education, due to the integrative nature of CC, it is evident that none of the traditional disciplines are capable of housing CC comprehensively. Thus far, CC has not been recognized within education systems as a discipline on its own right. Therefore, its integration within the existing, predominantly discipline-based curricula, poses particular challenges.

When introducing new curriculum that works across discipline boundaries, two strategies tend to be followed. The first is when two or more disciplines merge into one cross-disciplinary subject (e.g., NTS (nature, technology and society) [50]. The new cross-disciplinary subject may become

subsumed under one of the traditional disciplines and occupy a new disciplinary space of its own. Alternatively a non-disciplinary space is created within the curriculum, which may be occupied in various ways by cross-disciplinary studies [23,49]. Regarding the integration of CC into the curriculum, the literature reports on two common ways: The first and more prevalent way of integrating CC into the curriculum, is to fragment CC and disperse the parts into various established disciplines [23,49]. Often in this approach, one or two disciplines will carry the main burden of CC education, e.g., science and geography [51]. The second, less common approach is to introduce CC into the non- disciplinary and cross-disciplinary spaces, where CC may occupy part (e.g., subtopic) or the whole of these spaces [23].

The first approach of CC fragmentation is often referred to in the literature as cross-curriculum integration. While this approach is widespread, the disintegration, dispersal and subsuming of CC under existing curriculum subjects, has often been criticised for jeopardising CC education [36,52–54]. Potential damaging risks of CC education fragmentation include: (i) The omission of critical linkages between the various pieces of information taught in different subjects [49]; (ii) lack of collaboration between the disciplinary teachers in regard to CC education [24]; (iii) teachers perceiving CC as an add-on, unrelated to their discipline and therefore not engaging with CC education in their unwillingness to transcend beyond the boundaries of their disciplines; (iv) teachers' lack of expertese [23,29,50,55–57]; and (v) inadequate time for covering CC appropriately within the already crowded disciplines space [23,48].

In some countries CC is taught within cross-disciplinary or non-disciplinary spaces within the curriculum. These approaches are often referred to as project-based, theme-based and phenomena-based curriculum structures. Schreiner et al. [23] describe the limitations of this approach as experienced in the Norwegian curriculum model. In the 1970s and 1980s Norway introduced an integrated natural and social sciences cross-curriculum subject in primary and lower secondary schools. However, it was found that due to issues with teachers' qualifications and attitudes, the science elements were often not addressed. To rectify this, in a later 1997 reform, science and environmental studies were removed from the combined subject. Similarly, in teacher education courses, the cross-disciplinary subject of 'nature, society and environment', was abandoned. Meanwhile, Finland introduced phenomena-based learning in its 2016 curriculum reform where the individual schools choose and plan curriculum for a different phenomenon each year. Through this approach, students are expected to construct holistic understanding of issues linked to their communities and interests [49]. While the new approach is still in its early days, Lehtonen et al. [49] have already pointed out some challenges, mainly in regard to teachers' abilities and willingness for collaboration and sharing of expertise.

Overall, our review reveals limited theorising in relation to CC curriculum integration. In addition, there is also a lack of evidence for successful implementation strategies for CC education. Arising from our standpoint on the nature of CC (as outlined earlier), we now suggest that conceptualising CC as an integrated body of knowledge, with its own typography of governing principles, may give rise to new and different options for CC integration, such as allocation of a new disciplinary space for CC within the curriculum.

Overall in the review of the literature we found a gap in the conceptualisation of the nature of CC. In addition there was a broad spectrum of perceptions in the literature regarding the scope and content of CC education, ranging from science-based, to an integrative humanistic and science approach, as well as personal and social growth-oriented frameworks. Finally the question of CC integration within the curriculum received limited attention and theorising within the literature. In what follows we outline our methodological approach for developing our theoretical framework and the subsequent application to the Victorian curriculum examination.

## 3. Methods

In this section we initially describe the methods applied in the theoretical part of the study, followed by the methods applied in the empirical part of the study.

*3.1. Method for Developing the Theoretical Framework for Analysis*

The theoretical framework developed to produce CC curriculum analysis tools for our empirical section of the study, consisted of developing: (i) A set of characteristics that may be regarded as constituting the nature of CC; and (ii) a scoping map of CC-associated content knowledge. Both frameworks are derived from qualitative analysis of literary sources. The validity of the developed frameworks can be assessed by 'the extent to which interpretations of data are warranted by the theories and evidence used [58] (p. 267). In this case the framework validity is derived from the credibility and scope of the chosen literary sources and our interpretations of these sources [59]. The process of development of this two-part framework is described in the following.

3.1.1. Method for Identifying Essential Characteristics of the Nature of CC

To analyse CC conceptualisation within the study designs, there was a need to identify essential characteristics that constitute the nature of CC. A process of thematic analysis of relevant literature was applied to identify essential CC characteristics. The specific literature used and the output characteristics are described in details in the Results section below. It is important to note that the proposed characterisations of the nature of CC, are not regarded by us as either inclusive or exclusive. In the context of this study, they were developed primarily as a framework for the subsequent curriculum analysis. They are also presented here for further deliberations that may deepen our shared understanding regarding the nature of CC. In the second part of this study, the four natures of CC characteristics were applied as a priori categories for comparing and analysing the conceptualisation of CC in the various curriculum study designs.

3.1.2. Method for Scoping CC Content Knowledge

To develop understanding of the comprehensiveness of CC addressed by the study designs, there was a need to develop a map outlining the scope of CC contents. Mapping the scope of CC allows comparison and evaluation of the existing CC contents in each study design, in relation to the expectations set by the scoping process. This scoping framework assists in answering questions such as: How do we know whether a student who learned about CC in accordance with the study design in a particular subject, is sufficiently informed about CC? How would we know if there are 'holes' left unaddressed in students' CC knowledge? In other words, the mere fact that a study design mentions aspects related to CC is insufficient to convince us that climate change is taught appropriately. The scoping map allows for precise identification of the type and extent of CC content incorporated and any possible areas left as 'holes' in CC content in the specified curriculum.

Two main data sources were used for informing the CC scoping process. The sources were: (i) Various IPCC reports, primarily the IPCC Synthesis Report [13]; and (ii) the UNESCO Course for Secondary Teachers on Climate Change Education for Sustainable Development [47]. The IPCC reports were helpful in determining the major topics that constitute CC. The UNESCO course assisted in providing an educational framework for organising the various CC topics. These sources were thematically analysed to develop a scoping table of CC contents. The process of analysis initially yielded long convoluted lists of topics and content knowledge items. These were categorised and re-categorised to form the scoping map as it appears in Table 1. The scoping table of CC contents was then applied to evaluate the scope of content knowledge prescribed in each of the study designs.

It is important to note here that our scoping table was developed for the purpose of giving us a guide to what comprehensive teaching of CC could look like to assist our analyses of how the Victorian curriculum is performing relative to this perception of comprehensiveness. We acknowledge that further dedicated studies are required to define a broadly accepted scoping framework for CC education content knowledge suitable for a senior curriculum.

### 3.2. Method of Analysis of the VCE Study Designs

Australia has a national curriculum from foundation to year 10. However, each Australian state and territory has the responsibility to determine its own assessment and certification specifications for the upper-secondary curriculum [1]. Having no unified national years 11 and 12 curriculum, the present study focuses on one state's curriculum, as a case study for examination. The Victorian Certificate of Education (VCE) curriculum is comprised of upper-secondary study designs, organised by subjects ('Studies'). Each VCE study design follows a set structure. The first chapter, the introduction, includes: Rationale, aims and structure and additional administrative parts, not relevant to this analysis. The chapter on assessment and reporting follows the Introduction then the specific content and key knowledge outcomes expected for the study are organised into four units of study (roughly equivalent to four semesters of study, over two final years of schooling). The units are titled according to their content theme, which are each then subdivided into one to three areas of study or sub-themes. Each area of study includes a short description, specification of outcomes and associated key knowledge and skills. The final part of each of the four units outlines the assessment requirements for the unit [60]. In addition to the study designs, the Victorian Curriculum and Assessment Authority (VCAA) provides for each subject, support materials for teachers. These 'Advice for Teachers' documents incorporate of various information and advice for teachers including: Introduction and administration; developing a program; and teaching and learning activities [60]. The VCE subject study designs and their associated 'advice to teachers' documentation formed the data set for analysis. These curriculum documents were analysed with regard to CC content knowledge and suggestions as to how to teach the content. In addition, we examined whether there was any advice concerning cross-curriculum integration, for building on prior CC knowledge and complementing CC knowledge through collaborations between subjects.

The process of analysis involved an initial stage of identifying study designs in which CC is present. Once these study designs were identified, further curriculum analysis focused only on these CC-inclusive subjects. In order to identify the study designs that unambiguously addressed CC topics, there was a need to develop appropriate selection criteria. The approach taken was to develop a set of key words or phrases that indicate by their presence in the curriculum that a topic is clearly and directly addressing CC. The key words needed to be terms that repeatedly appear in association with CC to the extent that they are inseparable from the topic. These are words that are essential in any text addressing CC. Thus it is assumed that any study design in which any of these key terms appears may be regarded as addressing CC, even if the term climate change itself does not appear in the study design.

Data sources for developing our key CC identification words consisted of the four volumes of the IPCC Fifth Assessment Report: Mitigation of Climate Change [61]; the Physical Science Basis [62]; Impacts, Adaptation and Vulnerability Part A [63]; and the Synthesis Report [13]. The IPCC reports are internationally acclaimed sources and may be regarded as highly reliable in relation to plotting the scope of CC. The volumes were scrutinised for terms closely associated with CC. The examination process involved attentive reading through the reports to identify terms that appear in close association with the term CC. This examination of the four IPCC volumes [13,61–63] elicited a long list of terms that appeared in association with CC. Through a process of elimination, only the key words that appeared across all four volumes of the report were retained to result in four key CC indicator terms. These are: Climate change, global warming, greenhouse gases and carbon dioxide. Through this process it was determined that these terms are essential and central to any document related to CC. In other words, there is no way to discuss climate change without having at least one of the four terms mentioned. However, in regard to the term carbon dioxide, although it is a key term in every climate change discussion, unlike the other terms, it can also be used in other contexts (such as in teaching photosynthesis). It was therefore decided that the presence of this term on its own is insufficient for identifying a curricular climate change topic and it needs to appear in conjunction with one of the other three key terms.

The four identified key terms were used for identifying CC education within the VCE study designs. There are 94 VCE study designs, with 48 of these specifying a curriculum for various languages and as such are primarily language competency, not content, focused. The 46 non-language subjects are grouped into 11 disciplines and one investigation-based study. These 46 study designs were read through in search of our key CC indicator terms. CC was found within 10 of the study designs, namely: Australian and global politics [64]; environmental science [65]; physics [66]; economics [67]; agricultural and horticultural studies [68]; geography [69]; systems engineering [70]; chemistry [71]; outdoor and environmental studies [72]; and food studies [73]. Once the CC-inclusive study designs were identified, these ten study design documents formed the data set that were analysed in regard to conceptualisation, content knowledge and CC integration within the curriculum through the following processes. The data analysis for each of the study designs is presented as Supplementary Materials, available online.

CC conceptualisation within the study designs was assessed qualitatively by applying content analysis and discourse analysis. The aim of the content analysis was to ascertain the presence of the four natures of CC characteristics in each study design. Additionally, we examined emerging conceptualisations within the text, with the aim of understanding how the study designs themselves conceptualise CC. The discourse analysis was applied to identify institutionalised patterns of knowledge production. The findings across the ten study designs were summarised in a comparison table, in which the presence or absence of each nature of CC characteristics is signified by plus or minus signs (see Table 2).

The CC content analysis involved a process of systematic comparisons between each study design's content and the CC scoping map (see Table 1). The extent of comprehensiveness of CC content coverage in each study design was then rated by two authors of this paper, on a scale of 0–3 (least to most comprehensive). For each study design, the comprehensiveness of each CC theme within the study was rated separately and the scores were then added to form a comprehensiveness measure for each study design. Each of the comprehensiveness scores allocated to the study designs for each theme was averaged across the raters to produce the outcome scores. Inter-rater agreement was measured using Cohen's Kappa coefficient (k) [74], a reliability measurement that corrects for chance agreement between raters. There was substantial agreement between the two raters [74], k = 0.704, $p < 0.001$.

## 4. Research Outputs

The research outputs are organised into Parts A and B. Part A presents the theoretical frameworks developed in this study, including conceptualisation of the nature of CC and CC contents scoping map. Part B presents the results of the analysis of the Victorian Curriculum Years 10–11 study designs documents.

### 4.1. Part A: The Developed Theoretical Frameworks for Curriculum Analysis

#### 4.1.1. Characterising the Nature of Climate Change

Examination of the literature reveal some conceptualisations that are helpful in developing our understanding regarding the *nature of CC*. For example, Andrey and Mortsch [43] listed some characteristics of CC, which present challenges for education. These include: (i) Complexity; (ii) uncertainty; (iii) disproportionate impacts across countries and intergenerationally; and (iv) CC causes as embedded in current and preferred lifestyles [in 23] (p. 9). In addressing these characteristics, Schreiner et al. [23] presented an additional five characteristics of: (v) CC as a media issue; (vi) the invisibility of the problem; (vii) long time-scales suggesting that the consequences of today's behavior will be borne by subsequent generations; (viii) perceptions of CC as not a personal responsibility; (ix) competing environmental and political interests; and (x) perceptions of individual contributions to the problem as insignificant (pp. 9–10). While some of these characteristics may be regarded as inherent to the nature of CC (e.g., complexity and uncertainty), their intent is to describe the inherent challenges to educating about CC, rather than to characterise the inherent nature of CC as a body of knowledge.

Another form of CC conceptualisation was presented by González-Gaudiano and Meira-Cartea [44], where they categorised dimensions of CC through the structural nature of the problem. They identify structural barriers to change within categories of: (i) Those derived from the complex nature of the problem; (ii) those emerging from the moral and sociopolitical implications; and (iii) those related to the psychosocial and cognitive processes that condition the representation of climate change (p. 18). While these characteristics are helpful in addressing the difficulties in solving the problem, they too are not designed for characterising the nature of CC as a body of knowledge.

Other references to CC conceptualisation often appear as criticisms of perceptions propagated by the media and other interest bodies. For example, Selby [40] criticises the portrayal of CC as a problem of management, technological efficiency and/or responsible citizenship. Such perceptions give the erroneous impression that the $CO_2$ problem can be fixed within present terms of reference, rather than being understood as a crisis of humanity. Selby further criticises the advent of the sustainability education concept, suggesting it equates to offering the disease as a cure. Overall, the examples provided above highlight the need for more specific characterisation of the nature of CC.

For addressing this gap, a thematic analysis of a range of relevant literature was carried out for identifying consistent characterisations of CC. The analysis revealed a set of four natures of CC characteristics. These are: (i) CC is complex and involves multiple systems interactions; (ii) the study of CC involves cross (multi-inter-trans) disciplinary approaches; (iii) it inherently involves human action; and (iv) it involves a level of uncertainty. These characteristics appeared consistently in literature discussing characteristics of CC, as elaborated in the following.

CC as Complex Multi-Systems Interactions

Complexity and multi-systems interactions are frequently mentioned in most publications related to CC education (e.g., [23,29,40,44,45]). The U.S. Global Change Research Program publication entitled 'Climate change literacy: The essential principles of climate sciences' highlights the inherent complexity in its opening sentence that states: 'Throughout its history, Earth's climate has varied, reflecting the complex interactions and dependencies of the solar, oceanic, terrestrial, atmospheric and living components that make up planet Earth's systems' [75] (p. 1). Furthermore, the complex nature of CC is expressed under Principal 2 as follows: 'The interconnectedness of Earth's systems means that a significant change in any one component of the climate system can influence the equilibrium of the entire Earth system' [75] (Principal 2). Varying complexities permeate CC literature. Complexities such as the relationships between carbon dioxide emissions and CC indicators and observations; and the inherent socio–political–economic complexities associated with mitigation and adaptations to CC, encompassing connections among human health, water, energy, land use and biodiversity [13] (p. 31). These intricate interdependencies between the various systems constituting CC underlie Shepardson et al.'s [45] climate system framework developed for teaching about CC. The framework is organised around three essential questions, all related to complexity and multiple systems interactions (p. 330).

CC Involves Cross (Multi–Inter–Trans) Disciplinary Approaches

The most apparent characteristic of the nature of CC is the multi-inter-trans disciplinary approaches required for dealing with CC [39,44,48,49,76–78]. Bacon et al. describe the diverse disciplinary skills required for teaching and learning CC. These include:

(i) Fundamental physical sciences, social sciences and math needed for environmental assessment and engineering; (ii) basic economics including input-output analysis; (iii) industrial ecology and design at the process, plant, corporate, regional, national and global scales; (iv) information technologies for real time monitoring of processes, remote sensing of the environment and graphical information systems; (v) human and environmental impact modelling and risk assessment; (vi) social and behavioral research tools; (vii) understanding sustainability issues in a global context, with emphasis on the developing world; and (viii) professional and K-through-12 educational programs [56]. (p. 195)

Choi and Pac propose the following definitions for multi- inter- and trans-disciplinary. Multidisciplinary 'draws on knowledge from different disciplines but stays within their boundaries' [79] (p. 351). Interdisciplinary 'analyzes, synthesizes and harmonizes links between disciplines into a coordinated and coherent whole' [79] (p. 351) and transdisciplinary 'integrates the natural, social and health sciences in a humanities context and transcends their traditional boundaries' [79] (p. 351). These three types of disciplinary relationship appear consistently in various degrees in most published CC education literature and curricula materials, indicating all three integration perspectives are important within CC education (e.g., [39,44,48,49]. The term cross-disciplinary is used as a general term referring to any activity involving two or more academic disciplines [80] and we use this term here to encompass any multi-inter-trans disciplinary approaches.

CC Inherently Involves Human Action

It seems almost self-explanatory that CC inextricably involves human action. Climate change is a human induced phenomenon, driven by the current economic development model and population growth. Abatement of CC depends on collective human actions in order to change course and to limit adverse impacts [23,40,48,81]. The significance and urgency of the need for human actions is stressed repeatedly in the IPCC reports and in most CC publications [13,46,56]. For example, in the 2014 IPCC Synthesis Report, Summary for Policymakers (SPM), the following statements demonstrate the centrality and importance of collective actions by governments and decision makers at every level of society.

> Many adaptation and mitigation options can help address climate change, but no single option is sufficient by itself. Effective implementation depends on policies and cooperation at all scales and can be enhanced through integrated responses that link adaptation and mitigation with other societal objectives [13]. (p. 26)

> Climate change is a threat to sustainable development. Nonetheless, there are many opportunities to link mitigation, adaptation and the pursuit of other societal objectives through integrated responses (high confidence). Successful implementation relies on relevant tools, suitable governance structures and enhanced capacity to respond [13]. (p. 31)

Unlike any other curriculum discipline, CC can only be understood meaningfully in relation to the consequences of action or inaction in addressing CC by governments, policy makers, business sectors and communities.

CC Involves a Level of Uncertainty

Studies frequently mention future uncertainty in relation to CC, e.g., [39,44,45,48]. However, within the public debate over CC, two types of uncertainties often arise, with much confusion in regard to the differences between them. The first type of uncertainty related to public's scepticism regarding the validity of CC projections [82]. The second type is the uncertainty inherent to the nature of CC and its future projections. The following case illustrates this confusion.

The end of 2018 and early 2019 saw the catastrophic collapse of the Murray Darling Basin ecosystem, in south-eastern Australia, with over a million fish dying of suffocation [83]. Parallel to these events, a South Australia, Murray Darling Basin Royal Commission report [84] was published, pointing to the lack of incorporation of CC projections into the Basin Plan. One of the testimonies given by a representative of the Murray Darling Basin Authority (MDBA), provided the following explanation for the lack of incorporation of CC projections:

> One of the reasons proffered by the MDBA for postponing the incorporation of climate change projections into the Basin Plan in any meaningful way is that the science around it—and the consequent projections—are not certain [84]. (p. 250)

The Royal Commission found the failure of the MDBA to deal with CC future projections as 'indefensible' [84] (p. 247). When counteracting the above testimony and arguments related to uncertainty, the Commission explains that

> projections exist within a range creates only a level of uncertainty as to how much the Southern Basin will warm, and how much it will dry. That it will be both significantly warmer and drier is unfortunately not uncertain in any realistic sense. Further, the best available scientific knowledge often involves a best available estimate. Scientific analysis does not always and even often, result in absolutes. A climatic change projection is just that —it is unlikely to ever involve a statement that the climate will warm by a precise amount expressed in Fahrenheit or Celsius [84]. (p. 251)

The Murray Darling Basin catastrophe vividly illustrates how the MDBA adopted the public scepticism in its approach to CC projections, rather than integrating into its management plans the inherent uncertainties arising from the science of CC.

Deser et al. describe three sources of uncertainty related to future climate change. The first source of uncertainty is forcing. It 'arises from incomplete knowledge of the factors influencing the climate system', such as future trajectories of GHG emissions [85] (p. 527). The second source, model uncertainty, relates to differences in algorithms used in the various models, thus causing different models to yield different future predictions [85] (p. 527). Finally, the third source, internal variability, relates to 'natural variability of the climate system that occurs in the absence of external forcing and includes processes intrinsic to the atmosphere, the ocean and the coupled ocean-atmosphere system' [85] (p. 527). These three forms of uncertainty may be regarded as essential for developing meaningful understanding of CC future projections and thus form an inherent characteristic of the nature of CC. Whereas the first type, relating to public scepticism, is not inherent to the nature of CC in the same way that scepticism regarding the Theory of Evolution is not inherent to the nature of science.

### 4.1.2. Developing Climate Change Content Scoping Map

The data sources outlined above, under the 'Methods' section, were analysed and an extensive list of CC contents were derived from the literature. The initial list of contents was further organised through a process of categorising and re-categorising, to finally form two major categories of perspectives. These are: (i) Science dacts; and (ii) humanity: Socio-economic-political structures, networks, ethics and conduct. Under these two broad perspectives, the following key CC themes were placed on a continuum ranging from more science facts-based, to more humanity-based (and less science-based) aspects of CC. The themes are: Observed changes in the climate; drivers of CC; future CC; risks and impacts; adaptation and mitigation; socio-economic; policy and governance; and ethics. As typical of multi-system highly complex issues, none of these themes is exclusive and content knowledge topics may flow over from one theme to the other.

Finally the body of information from the thematic analysis of the literature was organised by: Fundamental questions and essential content knowledge. The essential content knowledge, was determined by allocating the relevant content items under the appropriate themes. Once the list of themes and their relevant content items was formed, we formulated the fundamental questions for each theme. This was done simply, by thoroughly reading through the contents of each theme and identifying the underlying questions, answered by the themes' contents. The developed Fundamental Questions serves the purpose of assisting the readers, by creating anchor points for the subsequent contents of each theme. This final process of formulating the fundamental questions and the essential content knowledge for each theme culminated our Climate Change Content Scoping Map, as presented in Table 1. This mapping of CC content knowledge was used as an evaluation framework for examining the CC contents present in the ten VCE study designs in Table 2.

**Table 1.** Climate change scope of contents: Perspectives continuum and content themes, ranging from science-facts-based to humanity-based, by fundamental questions and essential content knowledge.

| | Science Facts | | | | | Humanity: Socio-Economic-Political Structures, Networks, Ethics and Conduct | | |
|---|---|---|---|---|---|---|---|---|
| | *Observed Changes in Climate* | *Drivers of CC* | *Future CC* | *Risks and Impacts* | *Adaptation and Mitigation* | *Socio-Economic* | *Policy and Governance* | *Ethics* |
| **Fundamental questions** | What is climate and climate change? What are the instruments and means for measuring the climate in different time scales? What are the observed facts? (This aspect may be taught through an historical perspective tracking the path of data accumulation). | What causes CC? | How are future projections produced? What are CC scenarios? What are the sources of uncertainties in CC projections? What are the future projections of CC? | What are the risks and impacts posed by CC? What characterises risks and impacts distribution? | What are the roles of mitigation and adaptation? What are the means of mitigation? What are the means of adaptation? | What socio-economic processes drive and are impacted by CC? | What is the role of policy? What international, regional and national organizations, agreements and mechanisms are established for dealing with CC? | What is the role of ethics in combating CC? What are some of the relevant ethical dilemmas? |

**Table 1.** *Cont.*

| | Science Facts | | Continuum | | | Humanity: Socio-Economic-Political Structures, Networks, Ethics and Conduct | | |
|---|---|---|---|---|---|---|---|---|
| | *Observed Changes in Climate* | *Drivers of CC* | *Future CC* | *Risks and Impacts* | *Adaptation and Mitigation* | *Socio-Economic* | *Policy and Governance* | *Ethics* |
| **Essential content knowledge** | Explaining climate and climate change. Climate is the average weather in a given area over a lengthy period of time [86]. Climate describes the state of the atmosphere, influenced by the oceans, land surfaces and ice sheets. Climate change is a change in the statistical properties of the climate system that persists for several decades or longer—usually at least 30 years' [87]. CC data collection sources and methods of analysis include: Ice cores drawn from Greenland, Antarctica, and tropical mountain glaciers; tree rings; ocean sediments; coral reefs; and layers of sedimentary rocks. This ancient evidence reveals that current warming is occurring roughly ten times faster than the average rate of ice-age-recovery warming [88]. Overall observed changes indicate that atmospheric, surface, and ocean warming is unprecedented over decades to millennia [13] (p. 4). | Drivers of CC. Economic growth and population growth drive anthropogenic greenhouse gas emissions growth. These in turn are the dominant cause of warming [13] (p. 4). | Future projections production. Complex models are applied for developing long-term projections of CC [89]. Future greenhouse gases emissions are determined by complex driving forces, including: demographic change, socio-economic development, and rate and direction of technological change [90]. What are scenarios? Scenarios are alternative images of the future used to analyze how the driving forces may influence future emission outcomes and to assess the associated uncertainties. [89,90]. IPCC's four Representative Concentration Pathways (RCP) scenarios include: stringent greenhouse gases mitigation (RCP2.6), two intermediate | CC risks and impacts. Risks to *physical systems*, include: rivers, coasts, diminished snow, ice, glaciers, and permafrost cover. *biological systems*, include: desertification, ecosystem losses, mass extinction and reduced biodiversity. *human and managed systems* include: increased fires, cyclone, tsunami, floods, drought, malnutrition, diseases spread, economic losses, mortality, and displacement [13,15]. | The roles of mitigation and adaptation. *Mitigation* consists of actions to limit the magnitude or rate of long-term global warming and its related effects. Effective mitigation requires near zero emissions of $CO_2$ and other greenhouse gases by the end of the century [13] (p. 20). *Adaptation* aims to offset CC effects by reducing the vulnerability of social and biological systems. However, there are limits to its effectiveness [13]. Means of mitigation involve enhancement of technology, behaviour, production and resource efficiency. It requires both upscaling zero-carbon emission electricity generation, as well as reducing demand for energy. Mitigation efforts are required in all sectors. For example: In the energy supply—use of renewables (wind, solar, bioenergy, geothermal, hydro, etc.). In transport—fuel switching to low-carbon fuels. In building—apply integrated renewable energy sources/In industry—use of waste and carbon dioxide capture and storage. In Agriculture, Forestry and Other Land Use—methane reduction through livestock, reforesting, changes in human diet. | Socio-economic processes. Continued economic growth and patterns of production, distribution and consumption drive CC [12]. Sustainable socio-economic development is fundamental to mitigation and adaptation. | The role of policy Projections of greenhouse gases emissions depend predominantly on socio-economic development and climate policy [13] (p. 8). Governments must play a major role in combating CC. Effective implementation of CC policy depends on cooperation at all scales, and integrated responses that link adaptation and mitigation with other societal objectives [13] (p. 26). | The role of ethics. Reversing the course of CC requires social transformation of individual and collective assumptions, beliefs, values and worldviews influencing CC responses [13] (p. 27). Ethical perspectives are inherently involved in evaluation of present trends and conceivably future scenarios [47]. |

**Table 1.** *Cont.*

| | Science Facts | Continuum | | | | Humanity: Socio-Economic-Political Structures, Networks, Ethics and Conduct | | |
|---|---|---|---|---|---|---|---|---|
| | *Observed Changes in Climate* | *Drivers of CC* | *Future CC* | *Risks and Impacts* | *Adaptation and Mitigation* | *Socio-Economic* | *Policy and Governance* | *Ethics* |
| | Recent anthropogenic greenhouse gases emissions are the highest in history [13] (p. 2). Sphere-specific observed changes in *atmosphere*: Enhanced greenhouse effect; carbon cycle disturbances; increase weather variability; precipitation changes; and cloud cover changes [47]. *ocean*: Changes in ocean temperature and acidification; ocean circulation upheaval; coral bleaching; and changes in marine food chains [13,47]. *Land*: *Land cover:* Glacier melting; reductions in lake and river ice, soil moisture and runoff, and, permafrost cover [13,47]. *Land biomass:* Massive extinction of species; and early flowering [47]. Observed changes in extreme weather events: Extremes in warm temperature; high sea levels; and heavy precipitation [13] (p. 7). | Greenhouse gases are produced through: fossil fuel burning (energy production, industry, transportation); and, land use changes (urbanization, deforestation, agriculture) [13]. | (RCP4.5 and RCP6.0) and very high emissions (RCP8.5) [13] (p. 8). The meaning of uncertainty. CC projections are uncertain for the following three reasons: (i) they depend on scenarios of future anthropogenic and natural forcings that are uncertain; (ii) incomplete understanding and imprecise models; and, (iii) the internal climate variability [89] (p. 1034). Future projections of CC. Changes in surface temperature are projected to rise over the 21st century under all assessed emission scenarios [13] (p. 10). Intensification of projected extreme events includes: more frequent heat waves, lasting longer; more intense and frequent extreme precipitation; continuing warming and acidification of the ocean; global sea level rise [13] (p. 10). The risks of abrupt or irreversible changes increase, as the magnitude of the warming increases. Components of the climate system will undergo long lasting changes [13] (p. 16). | Risks and impacts distribution. Risks are distributed unevenly. The most disadvantaged people are most vulnerable to be strongly affected by CC. Poor countries are more vulnerable than rich countries [13]. The risks and impacts vary by geographic regions. For example, some regions are more at risk of wildfires and extreme heat, while others are at risks of floods [13]. | Means of adaptation require coordinated actions in 10 categories: (1) Human development (such as improved education, health, and nutrition); (2) poverty alleviation; (3) livelihood security; (4) disaster risk management (such as early warning systems); (5) ecosystem management (such as urban green spaces); (6) spatial or land-use planning (such as provisioning of adequate housing); (7) structural/physical adaptations in regard to engineering and built environment, technology, ecosystem-based options, and services; (8) institutional adaptations, including: economic options, law and regulation and, national and government policy and programs; (9) social adaptations including educational, informational and behavioural options; (10) spheres of change include: practical, political and personal [13] (p. 27) | Socio-economic processes. Continued economic growth and patterns of production, distribution and consumption drive CC [12]. Sustainable socio-economic development is fundamental to mitigation and adaptation. Climate change processes and impacts involve globalization, increased socio-economic inequality; inequality in access to resources; unfair distribution of CC risks; increased vulnerability and reduced resilience; urbanization; rural and urban poverty; gender inequality; displacement; conflict and economic refugees; health impacts, including spread of infectious diseases, malnutrition and respiratory diseases; and mortality [13,15]. | Climate change processes and impacts involve globalization, increased socio-economic inequality; inequality in access to resources; unfair distribution of CC risks; increased vulnerability and reduced resilience; urbanization; rural and urban poverty; gender inequality; displacement; conflict and economic refugees; health impacts, including spread of infectious diseases, malnutrition and respiratory diseases; and mortality [13,15]. | International, regional and national organizations, agreements, and mechanisms developed for dealing with CC, including: The United Nations Framework Convention on Climate Change (established in 1994) and the yearly Conference of the Parties (COP); the Paris Agreement; the Inter-governmental Panel on Climate Change (IPCC) and its role in assessing the scientific, technical and socio-economic information relevant for the understanding of the risk of human-induced CC; UNESCO's role in CCE. The role and actions of each national policy and governance in regard to CC [16–19]. | Relevant CC ethical issues. Some CC ethical issues include: Inter-generational justice and accountability; lifestyle choices; social justice and unfair distribution of risks; human rights and displacement; traditional lifestyle changes, such as risk to subsistence farming and fishing, vulnerability and resilience building [47]. |

**Table 2.** Climate change conceptualization, by the ten Victorian Certificate of Education (VCE) subjects, the nature of climate change, perceptions and misconceptions.

| VCE Study Designs | Nature of CC | | | | Overall Perceptions | Misconceptions or Misalignment with Conventions |
|---|---|---|---|---|---|---|
| | Complexity and Multiple Systems Interactions | Cross-disciplinary Approaches | Inherently Involves Human Action | Involves a Level of Uncertainty | | |
| Australian & Global Politics | + | + | + | - | CC is a **human crisis** | |
| Environmental Science | + | + | + | - | CC is an outcome, requiring behavioral, ethical and technological responses | • The enhanced greenhouse effect perceived as a cause of CC rather than forms part of CC processes. • Astronomical and solar systems presented as responsible for CC |
| Physics | + | + | - | ± | | Reference to CC as climate science or enhanced greenhouse effect |
| Economics | - | + | + | - | CC is an outcome of economic growth | |
| Agricultural and Horticultural Studies | + | - | + | - | CC is primarily a **problem of management** | |
| Geography | + | + | - | - | CC is a **cause, a process, an outcome**. It is human induced | |
| Systems Engineering | + | + | - | - | CC is a problem that requires **technological fix** | |
| Chemistry | - | + | - | - | CC is a problem of technological efficiency | |
| Outdoor and Environmental Studies | - | - | - | - | CC is a cause | • CC is a natural change • It is a question of debate whether humans caused CC or it is a natural process. |
| Food Studies | + | - | - | + | CC is an outcome state forming environmental risk | |

+ indicates characteristic present; - indicates characteristic absent; ± indicates partial presence.

*4.2. Part B: Results Analysis of Climate Change Education within the VCE Study Designs*

The empirical findings from the analysis of the VCE Study Design curriculum documents are organised according to the three aspects of CC education applied for the analysis: CC conceptualization, scope of content knowledge and cross-curriculum integration. The Part B findings presentation is followed by a discussion of the findings in the subsequent section of the paper.

4.2.1. CC Conceptualisation across the Study Designs

The findings regarding CC conceptualisations are summarised in Table 2. The table shows the presence of the four characteristics of the nature of CC within the study designs. The presence or absence of the four natures of CC characteristics is signified by plus or minus signs. The plus sign signifies presence in the study design and the minus-sign signifies absence. A plus and minus combined, signifies partial presence. In addition, in the last two columns of the table, we present how the study designs themselves conceptualise CC and whether or not there were apparent misconceptions.

Examination of Table 2 reveals that none of the VCE study designs presents a complete conceptualisation of the nature of CC. Australian and global politics and environmental science represent three of the four characteristics. Another six study designs represent two characteristics. Chemistry represents one and outdoor and environmental studies has no representation. While most study designs represent the first two characteristics of CC, complexity and multiple systems interactions and cross-disciplinary approaches, the latter two characteristics are under-represented. Only Food Studies presented the characteristic involves a level of uncertainty. Physics did so to a limited extent.

Most study designs present CC conceptualisation as either a cause or as an outcome. Geography is distinguished in that CC is presented as all three—cause, process and outcome. In the food studies study design, the outcome is narrowed to a risk factor to the environment.

Another type of CC conceptualisation that is evident is that of CC as a problem. Study designs differ in their answer to the question: A problem of what? Agricultural and horticultural studies presents CC as a problem of management, while System Engineering and Chemistry presents CC as a problem of technology. Australian and global politics stands out in their presentation of CC as a crisis of humanity.

A particularly worrying finding is that two study designs present clear misunderstandings regarding CC. Environmental science presents two misconceptions. The first is the perception that astronomical and solar systems are responsible for CC. The second relates to the perception that the enhanced greenhouse effect causes CC, rather than it forming part of CC processes. As Table 1, clearly demonstrates, the causes of CC (which appear under the theme 'drivers of CC') are economic and population growth and not the enhanced greenhouse effect, which is one of the observed outcomes. Outdoor and Environmental Studies presents CC as a natural change and it is debatable whether humans are causing it. In addition, CC is also narrowly conceptualised, as a cause of environmental degradation.

4.2.2. The Scope of Content Knowledge across the Study Designs

The findings regarding CC content knowledge addressed by the VCE study designs is summarised in Table 3, using a scoring method. The scores are derived from the analysis of each study design in comparison with the scope of CC content mapped in Table 1. The level of comprehensiveness of the content knowledge in each theme is indicated by a score ranging 0–3, where, 0 = no content; 1 = minimal content; 2 = medium amount of content; 3 = comprehensive content. The score for each theme was summed across the eight themes and a cumulative score given to each VCE subject. The maximum possible score is 24 points.

**Table 3.** Levels of comprehensiveness of Climate Change (CC) themes addressed by the ten Victorian Certificate of Education (VCE) subjects.

| VCE Subjects | Science Facts | Continuum | | | | Humanity: Socio-Economic-Politic Structures, Networks, Ethics and Conduct | | | Sum of Scores * |
|---|---|---|---|---|---|---|---|---|---|
| | *Observed Changes in Climate* | *Drivers of CC* | *Future CC* | *Risks and Impacts* | *Adaptation and Mitigation* | *Socio-Economic* | *Policy and Governance* | *Ethics* | |
| Australian & Global Politics | 2 | 3 | 1 | 2 | 3 | 3 | 3 | 0 | 17 |
| Environmental Science | 3 | 1 | 1 | 1 | 2 | 0 | 0 | 0 | 8 |
| Physics | 2 | 1 | 1 | 0 | 0 | 0 | 0 | 0 | 4 |
| Economics | 0 | 2 | 0 | 0 | 1 | 0 | 1 | 0 | 4 |
| Agricultural and Horticultural Studies | 0 | 1 | 0 | 1 | 1 | 0 | 0 | 0 | 3 |
| Geography | 1 | 0 | 0 | 1 | 1 | 0 | 0 | 0 | 3 |
| Systems Engineering | 0 | 0 | 0 | 0 | 1 | 0 | 0 | 0 | 1 |
| Chemistry | 0 | 0 | 0 | 0 | 1 | 0 | 0 | 0 | 1 |
| Outdoor and Environmental Studies | 0 | 0 | 0 | 0 | 0 | 0 | 1 | 0 | 1 |
| Food Studies | 0 | 0 | 0 | 0 | 0 | 0 | 0 | 0 | 0 |

* Comprehensiveness of content knowledge score: 0 = no content; 1 = minimal content; 2 = medium amount of content; 3 = comprehensive content.

Examination of Table 3 reveals that only the Australian and global politics study design addresses CC in a relatively comprehensive way, with a score of 17. Environmental science received less than half the points, scoring 8. All other subjects received scores ranging 0–4 points. The findings suggest that in only one out of 94 VCE subjects CC content knowledge is addressed at an acceptable level of comprehensiveness. However, in this single subject, the Australian and global politics, the unit about CC is offered as an elective and thus may not be taught in the delivery of this study (depending on teacher and student choices). Further examination of the CC content analysis reveals that none of the study designs address the ethical aspect of CC and overall, content knowledge tends toward the science-facts end of the perspectives continuum (see Table 1). CC humanity perspectives appear in the study designs primarily in regard to adaptation and mitigation.

In environmental science, the term climate change is conspicuous in its absence from the study design. On the rare occasions when it is mentioned, aspects related to CC seem to be concealed under the reductionist term enhanced greenhouse effect. This in turn is portrayed as a problem of an imbalance of gases in the atmosphere. The choice of terms begs querying the political underlying motives.

In the VCE physics study design, the term climate change is absent altogether. Instead, the study design refers to CC as climate science or enhanced greenhouse effect. This is in contrast to internationally accepted scientific terminology. The IPCC refers to the term climate change as the broad phenomena encompassing the eight CC content knowledge themes used as organising principles in our mapping in Table 1 [12]. Whereas climate science is the scientific discipline of climatology, which focuses on the study of the Earth's weather patterns and the systems that cause them [91].

Overall, in eight of the ten study designs under analysis, the level of CC content is poor, even close to non-existent. The findings lead to the conclusion that the vast majority of years 11 and 12 graduates in the state of Victoria, Australia, who were taught in accordance to the 94 study designs, are not well educated about CC in their upper-secondary studies.

### 4.2.3. Cross-Curriculum Integration Across the Study Designs

The aspect of cross-curriculum integration was examined in each study design. None of the study designs address cross-curriculum integration in any way. The various study designs seem to operate in silos in regard to CC education. We acknowledge that the discipline specialists involved in developing the various study designs may not have been directed to address cross-curricular CC or even sustainability perspectives, as this priority is only indicated in curriculum to year 10 in Australia. When CC was addressed in a study design this was usually just in the context of the subject/discipline, without the broader, interconnected issues addressed and no provisions made for complementing them elsewhere. Some study designs seem to rely on an unsubstantiated assumption that students have prior knowledge in CC. For example, in the economics study design students are requested to write a report to the minister with policy recommendations regarding 'tackling' CC [92] (p. 17). It seems obvious that addressing such a question requires substantial amount of CC knowledge, which is not addressed by the economics study design. These types of suggested learning activities, which are abundant throughout the VCE, raise the question: Where else in their studies, can students obtain the required prior knowledge about CC? The VCE study designs do not provide answers to this question, nor provisions for complementing students' existing CC knowledge.

### 5. Discussion

The objectives of this study are to develop a suite of theoretical tools for evaluating CC curricula, and to apply these frameworks for examining: the nature of CC conceptualisation, the comprehensiveness of CC contents, and the integration of CC, within the Victorian upper-secondary curriculum. In regard to CC conceptualisation, the findings reveal that all 10 study designs analysed present incomplete representation of the *nature of CC*, often focusing on one narrow aspect of CC. A troubling finding is the identification of CC misconceptions in two subjects, Environmental Science, and Outdoor and Environmental Studies. Also worrying is the fact that Physics avoids application of the internationally

accepted term climate change and uses the term climate science instead. This tendency to avoid using the term CC, regardless of its appropriateness to the context of the text, was found throughout most of the other study designs to a lesser extent. In regard to CC content knowledge, the findings reveal that only one study design, Australian and Global Politics, out of the 94 offered, presents a close to holistic and comprehensive presentation of CC. However CC is offered in this subject only as an elective. In addition, no evidence was found for cross-curriculum integration and collaboration, as would be expected with a holistic implementation strategy. These finding suggests that, on the whole, graduates of the Victorian Certificate of Education are not educated about CC.

In what follows, we discuss these findings with regard to three major implementation gaps reflected in the findings: between the *nature of CC* and its representation within the curriculum; between the scope of CC content knowledge themes as understood by the scientific community, and the scope of CC present in the curriculum; and, between what may be regarded as 'best practice' curriculum integration and the existing approach to integration. In addition we discuss the role of Environmental Science in promoting CC. The implications of the study findings are discussed, where relevant, within each discussion section.

### 5.1. Gap between the Nature of Climate Change and its Representation within the Curriculum

Climate change is widely accepted as the most pressing crisis of our time, posing unprecedented challenges to humanity [2,3]. CC education models such as UNESCO's CC teaching model for secondary teachers, and Tolppanen et al.'s [48] Bicycle CC education model emphasize CC as a human existential problem, rather than as a scientific technological problem [47,48]. Yet our findings reveal that only Australian and Global Politics presents CC as a human crisis. Most other study designs present CC in a reduced form: as an outcome, cause or technological or managerial problem.

Two study designs not only discount the scope and acuteness of CC, but also present factual misconceptions. Environmental Science presents misunderstanding of scientific facts. Outdoor and Environmental Studies presents a misconception, assuming that CC is a natural change and it is debatable whether humans are causing it. To our view it is unacceptable to debate the question of whether humans cause CC in a VCE study design. This question has been firmly resolved by the scientific community and it is beyond debate, similar to the none-debatedness of the Theory of Evolution versus Creationism. In addition, Outdoor and Environmental Studies incorrectly addresses the concept of CC uncertainties. As described earlier, public scepticism regarding CC may sometimes be confused with CC projections' uncertainties, inherent to the *nature of CC* [82]. Outdoor and Environmental Studies present conceptualisation of uncertainty of the first type, rather than of the second.

Overall, the findings suggest that a fundamental conceptualisation gap prevails throughout the study designs in regard to the *nature of CC*. These lags may be attributed to any of the causes suggested by Foppoli, et al. [6], which include lack of materials, teacher training and public support. In the case of CC conceptualisation, we suggest a fourth explanation related to the Australian political climate surrounding the topic. It seems plausible to suggest that the minimisation of the CC phenomena within the curriculum could be attributed to alignment with the government of Australia's prevailing political positioning of limiting climate solutions.

The implication of the findings regarding the conceptualisation gap, is that there is a need for robust discussion and engagement between curriculum developers, educators, scientists and the public in relation to the *nature of CC* and how it should be represented in school curricula. These discussions may hopefully lead to development of a shared understanding regarding the *nature of CC*, and in turn pave the way to more rigorous implementation of CC education within text books, resources, teaching and learning.

### 5.2. Gap between the Scope of Climate Change Content Themes and their Representation in the Curriculum

Curriculum implementation lags often present similar characteristics in different countries. Our findings in the Australian context, suggest that when CC content knowledge themes do appear in

the curriculum, this occurs mostly in an anecdotal, disorganised and incoherent way; often emphasising only one or two themes and neglecting the others. Plutzer, et al. [93] in the U.S. found similar results, when addressing the question of CC content knowledge, from a science teachers' perspective.

The findings reveal that the theme adaptation and mitigation is addressed more extensively compared to other themes, while ethics is not addressed at all in the context of CC (although addressed in other contexts). Mitigation and adaptation inherently involve ethical issues of equity and justice, as those who are most vulnerable to CC contribute the least to GHG emissions [13] (p. 17). Since ethical decisions are at the heart of mitigation and adaptation, this finding raises a question regarding the depth of learning regarding mitigation and adaptation within the VCE study designs. The findings also suggest that the theme adaptation and mitigation is predominantly addressed from a technical perspective, rather than from a human perspective. Supporting this interpretation is the fact that the Humanity perspectives are under-represented in most study designs (see Table 3). These findings suggest that there is a need to invest efforts in creating better connections and integration across CC themes, and enhancing the human perspectives in CC education within the VCE study designs. These dual efforts of connecting across CC themes, and enhancing the Humanity perspectives, should be an area of focus for future CC curriculum development.

### 5.3. Gap between 'Best Practice' Curriculum Integration and the Existing Approach to Integration

The findings suggest that cross-curriculum integration is the chosen approach for teaching CC in the upper-secondary Victorian curriculum, as there is no dedicated CC study design and CC topics are integrated into the discipline-based studies. Only 10 out of 94 subjects carry this burden, each to a very limited extent. Fogarty [94] describes ten ways to connect the curriculum at implementation levels ranging from 1 to 10. The findings suggest that the level of CC curriculum implementation in the upper-secondary Victorian curriculum most closely aligns with the basic Level 1, described as 'separate and distinct disciplines' (p. 61), where no provisions are made for higher levels of integration of concepts across the curriculum.

While cross-curriculum integration is often promoted in the literature as best-practice for CC education [23], some criticisms of the approach highlight potential risks. These include the risks of: critical information left out as 'holes' in content knowledge [40]; deficits in critical linkages between the various pieces of information [49]; and, insufficient time allocation within the subjects to cover the CC contents appropriately [23,25]. Our findings provide evidence for all three risks. In line with the above, we found major 'holes' in the CC content; lack of linkages between the fragmented pieces of information, dispersed across the study designs; and insufficient time allocation to address CC. In regard to time allocation, we draw this conclusion from the structural organisation of the curriculum. Each unit in a study design approximates to one semester of teaching. In all of the study designs CC was taught as segments within a unit, thus suggesting that none of the study designs allocated a full semester for teaching about CC. Evidence suggests that these curricular structural limitations flow through to implementation. In addition there is no curricular priority within the VCE to integrate across subjects so such integration efforts have to be instigated by the teachers. Bacon, et al. [56], when attempting to implement the cross-curriculum approach at a tertiary level, found major obstacles in teachers' willingness to collaborate across disciplines.

The approach to CC integration within the curriculum is closely related to its conceptualisation. The cross-curriculum approach to integration seems to form a barrier to conceptualising CC as a body of knowledge with typical *nature of CC* characteristics. The complexity of CC, and the inseparability of its parts, make it almost impossible to effectively teach CC through fragmentation. Our evidence clearly indicates that this approach is ineffective. Other approaches of CC integration trials have been described in the literature, such as teaching CC through theme-based, phenomena-based or project-based approaches. However there is no evidence for successes in states-wide curricular application of this CC integration strategy. More so, there is some evidence for lack of success, when these approaches were applied in the past in various contexts [23].

Overall, the evidence suggests that cross-curriculum integration is ineffective as an approach for teaching CC in the upper-secondary school level. We also suspect that new cross-disciplinary or non-disciplinary, inquiry-style spaces within the curriculum might not be as successful as hoped due to systemic CC education issues. Both integration approaches seem to suffer from similar shortcomings, including: poor conceptualisation of the *nature of CC*, lack of teacher expertise, and lack of cross-disciplinary collaborations either between the discipline silos, or within the cross/non-disciplinary spaces. Therefore we propose establishing a new curriculum discipline dedicated to CC education. This approach has the potential for multiple CC education gains by providing a framework for discussing CC curricula and the *nature of CC*; by supporting specialisation among teachers; and through developing evidence-based approaches for CC curriculum implementation. Overall, we maintain that the establishment of CC discipline within the curricula may open new pathways for improving the quality of CC teaching and learning for students.

### 5.4. The Role of Environmental Science in Promoting CC

No other curriculum subject is as close to CC as Environmental Science. If there is one subject that may be expected to take a leading role in educating about CC, this would be it. The importance of this subject in relation to CC education motivates us to dedicate a specific section to discussing the analysis results of the Environmental Science study design.

The findings reveal that contrary to the expectation of finding CC education leadership in this subject, the analysis suggests the opposite. VCE Environmental Science seems to be critically lagging behind in many ways. This lag was found in regard to the three aspects of CC; conceptualisation, content knowledge and integration. The study design presents critical factual mistakes in CC conceptualisation. The level of content knowledge presented is poor, scattered and incoherent. Throughout the study design CC is consistently addressed in a narrow and reduced form, as an effect or an outcome that is being measured, often appearing disguised under the alternative term enhanced greenhouse effect. Here we wish to draw specific attention to the following error:

Area of Study 2 in Unit 4, with the peculiar title 'Is climate predictable?' (Evidently, the answer is both 'yes' and 'no'), describes the scope of learning as follows:

> In this area of study students investigate the astronomical, solar, and Earth systems and human-based factors that have altered important relationships between the energy, water and nutrient cycles, resulting in the enhanced greenhouse effect and climate change. They compare natural and enhanced greenhouse effects and their significance for sustaining ecological integrity [65]. (p. 30)

The phrasing of this sentence suggests that astronomical and solar systems are responsible for CC. This is an error, in contradiction with scientific facts. The NASA Global Climate Change Internet site answers the question: 'Is the Sun causing global warming?' unequivocally, as follows: 'No. The Sun can influence the Earth's climate, but it isn't responsible for the warming trend we've seen over the past few decades' [95]. In another NASA Internet page dated 3 January 2012 it is stated that 'A new NASA study underscores the fact that greenhouse gases generated by human activity - not changes in solar activity—are the primary force driving global warming' [95]. Unit 4 seems to disregard the IPCC reports, explaining the causes of CC as follows:

> Anthropogenic greenhouse gas emissions have increased since the pre-industrial era, driven largely by economic and population growth, and are now higher than ever. This has led to atmospheric concentrations of carbon dioxide, methane and nitrous oxide that are unprecedented in at least the last 800,000 years [13]. (p. 4)

The examination of the study design led us to query the quality of the document beyond its relation to CC education, as exemplified above. We couldn't ignore noting that the Environmental Science study design is scattered with concerning questions and statements that seem inappropriate in

a governmental statutory document of such high importance. There are many such examples and we present a few, as follows.

Under Area of Study 2, under Category 2, entitled 'Water pollution', appears the following question: 'Can the Great Barrier Reef be quarantined to alleviate coral bleaching?' [65] (p. 20). The question is pedagogically and informationally misleading because it directs students toward the wrong path in their conceptual development. The phrasing of the question presents an erroneous understanding regarding the causes of coral bleaching. It potentially suggests to the reader that coral bleaching is caused by some invasive species that may be blocked out through quarantining. This form of questioning obstructs both teachers and students from a pathway leading toward comprehension of accurate scientific facts. The facts are that coral bleaching is caused by environmental stressors, predominantly ocean warming and acidification as a consequence of CC; in particular, the raised level of atmospheric carbon dioxide and it's take up by the ocean [96].

Another disturbing question in this study design is: 'Has the global response to the hole in the ozone layer made a difference over time?' [65] (p. 20). It has long been established that the term *hole* may lead to the development of students' misconceptions, thinking that there is a physical hole in the ozone. It is now accepted that the term *ozone depletion* should be used rather than *hole* [97,98].

Another puzzling question appears in the title of Unit 3, Area Study 2, as follows: 'Is maintaining biodiversity worth a sustained effort?' [65] (p. 24). We think that questioning the importance of biodiversity conservation in the VCE Environmental Science study design is inappropriate. Such a question may be expected in the media, but it is concerning when appearing in an official document that should be an exemplar for promotion of evidence-based scientific knowledge acquisition. We would have instead preferred to see open questions inviting deep system thinking regarding the causes and impacts of biodiversity loss and conservation.

In Area Study 2 of Unit 3, appears the troubling title: 'Is development sustainable?' [65] (p. 26). We ask if this is an appropriate question to ask at this time with compelling evidence of an imminent CC crisis? The answer to this question has already been given, and it is 'no'. We would have preferred instead, to see open questions inviting students to consider the costs of development.

Unit 4, Area of Study 1 is entitled: 'What is a sustainable mix of energy sources?' [65] (p. 29). We wonder if anyone is capable of providing an answer to such a question. Does the unit claim to do so? We caution against over simplification of complex problems. In the same area of study also appears the following paragraph:

> In this area of study students examine the concepts associated with the use of different forms of energy by human societies. Focus moves from understanding the relationship between the uses of local sources of energy to examining the global impacts of these uses, including consideration of the consequences over short (seconds to years), medium (multiple years to hundreds of years) and long (thousands to millions of years) time scales. Students investigate through field and practical activities the extent, availability, consequences, and alternative forms of energy available while considering the environmental, social and ethical challenges involved [65]. (p. 29)

Once again we question the underlying rational of this learning activity. In this activity students are requested to consider the global impact of using sources of energy in time scales ranging between seconds to millions of years. Standard categories of time scales used in similar contexts usually consider short term, as ranging from present to 15 to 30 years; medium term, as 15/30–40/50 years from present; and long term as approximately 60/70–90 years from present (based on IPCC synthesis report, 2014 [13]). The exact timeframes vary in accordance to context. However, no publication discusses energy use projections in time scales ranging from seconds to millions of years. This is simply nonsensical. More so, how could students possibly predict impacts of energy use a million years from now, or seconds from now? Both extremes of the timescale are inappropriate for this learning activity.

It is beyond the scope of this paper to present the full range of our reservations concerning the structure and content of this study design. However, the few examples provided might be an indication that it is not only CC that is inappropriately conceptualised in VCE Environmental Science.

The implications of our findings suggest that there is a need for an overhaul examination of the Victorian VCE Environmental Science study design' in regards to its approach to CC education. There is a particular need to reconsider the role of Environmental Science in enhancing CC education, at the current time, when cross-curriculum implementation is proving to be ineffective, and with no alternative strategies yet being considered. Environmental Science, if appropriately revised, may be strongly positioned to take a leadership role in the teaching and learning of climate change holistically.

## 6. Limitations and Further Studies

There are various limitations to the findings of this study. The study examined the upper-secondary study designs in the state of Victoria, Australia, and found the curriculum in terms of CC education to be lacking. The findings, however, are limited to only this state's curriculum. Further studies in other Australian states and other countries would assist in creating a broader knowledge-basis regarding what we consider as a generation-old implementation-lag of CC education. In particular, it would be helpful in inquiring into 'best practice' CC education to examine other implementation models and compare the various approaches, not only in regard to upper-secondary curricula, but also regarding the lower year levels of schooling.

## 7. Conclusions

Climate change can no longer be regarded as mere phenomena. Since the early publications by Fourier (1824) [10,11], CC arose as a body of knowledge forming a paradigm shift in our understanding of the critical implications of the relationships between the economy, society, global politics and the natural environment [39,40]. Historically, major paradigm shifts have been met with resistant curricula. Paradigm shifts are challenging because their integration into existing curricula requires more than just add-on, simple fixes. Often they require reconceptualisation and restructuring of either the entire curriculum, or parts of it. The new discipline of CC poses particular challenges to existing curricular frameworks. Not only can CC not be accommodated fully into existing discipline-based subjects, but challenges also arise due to conflicts between the underlying value system of CC and the existing socio-economic political structures. In the state of Victoria thus far these challenges seem to have been met by only tokenistic introduction of CC as addendum to the curriculum. Unlike Darwin and Einstein's theories that could wait for over a century for curricula reforms, CC education implementation is urgently needed and can wait no longer. The findings of this study highlight the need to overcome the implementation gap, through curricula reforms that are capable of accommodating the CC discipline in its whole. Effective implementation strategies need to be evidence-based.

In this study we mapped out key characteristics of the *nature of CC* and the content themes constituting CC to enable evaluation of CC education in the Victorian upper-secondary curriculum. In order to support further development of CC curricula, it is imperative that these two dimensions of CC be further discussed and researched to strengthen foundations for CC education. We offer our *nature of CC* characterisation, and the CC perspectives continuum (ranging between science-facts-based, and humanity-based) (see Table 1) as a prompts for such discussions. We identify a need in the field to develop a deeper understanding of what constitutes CC as a discipline, and curriculum models that reflect this understanding. CC curriculum development also needs to consider the appropriate year levels for studying CC, and what content scope is suitable for the given Year levels. Most of all, this study affirms the importance of developing a new model for integrating CC education within the secondary curriculum.

**Supplementary Materials:** The following are available online at http://www.mdpi.com/2071-1050/12/2/591/s1, Study Designs Detailed Analysis.

**Author Contributions:** Conceptualization, E.E. and H.W.Q.; Methodology, E.E. and V.P.; Formal Analysis, E.E; Investigation, E.E.; Resources, E.E. and V.P.; Writing—Original Draft Preparation, E.E.; Writing—Review & Editing, E.E. and H.W.Q.; Supervision, E.E. and H.W.Q. All authors have read and agreed to the published version of the manuscript.

**Funding:** This research received no external funding.

**Conflicts of Interest:** The authors declare no conflict of interest.

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
