# Peer review of "Climate Change Education: Mapping the Nature of Climate Change, the Content Knowledge and Examination of Enactment in Upper Secondary Victorian Curriculum"

_sustainability, doi:10.3390/su12020591_

Round 1

Reviewer 1 Report

Thank you for the opportunity to review this article. In this article the authors examine how the curriculum in Victoria addressess climate change in upper secondary school. The article examines an interesting and important issue on how climate change is taught and what are the potential short falls of climate change education in the examined curriculum.

Though the context of the study is somewhat interesting, I find that the article needs extensive work before it can be published. My three main concerns are (i) the authors familiarity with the existing literature, (ii) the way the study is framed and (iii) the reliability of their analysis. I will explain these below in furhter detail.

Existing reseach
The authors present the "nature of CC" as having four distinct characteristics, saying:

"The analysis of literary sources reveal at least four characteristics that appear consistently, and may be considered as characterising the nature of CC" (line 214-215)

However, the authors do not present an extensive literature review on the topic, nor do they refer to a review article where these four characterstics stem from. In fact, the literature shows that the nature of CC should be addressed much more broadly than what these four characteristics present. The authors should acknowledge the existing research and justify why they have decided to focus on only these four points as "nature of CC", rather than taking a broader view. For some reference of a broader view, please read:

Schreiner et al. (2005): Climate Education: Empowering Today's Youth to Meet Tomorrow's Challenges.
Cantell et al. (2019): Bicycle model on climate change education: presenting and evaluating a model.

In addition, the four points should acknowledge the existing research within these four characteristics. For instance, Shepardson has written extensively on how to address the scientific topics of CC. See e.g.

Shepardson et al. 2011. Conceptualizing climate change in the context of a climate system:implications for climate and environmental education.)

and Jensen et al. have written about the importance of action (=action competance)

(see e.g. Jensen & Schnack, 1997: The Action Competence Approach in Environmental Education.)

What concerns me is that on several occasions the authors make strong statements on the existing research, which are not accurate, and shows that the authors are not familiar with the extent of the existing research. For instance:

(line 89-90) "very little is known regarding CC education implementation in school curricula"

(line 133-134) "our search of the literature has found no systematic analysis of any national curriculum in regard to CC education implementation."

Statements like the ones above do not stand scrutiny. In fact, even the authors present some studies in which curricula have been studies (and many more exist). Discussion of curriculum is often embedded into articles, though a "systematic analysis" would not be their primary focus, so I don't know what exactly you mean by "systematic analysis", but these statements give the idea that curricula have not been examined, which is not the case.

Also lines 319-345 contain many bold statements, that do not do justice to the existing research. Many articles have specific and practical suggestions on how to address different aspect of CC, especially in their discussion. See e.g.

Shepardson et al. 2011. Conceptualizing climate change in the context of a climate system:implications for climate and environmental education.)

Tolppanen & Aksela. 2018. Identifying and addressing students' questions on climate change.

The way the study is framed
Authors make the claim that cross-curricula education gives a fragmented picture of CC, which is probably true. However, authors present a lack of theoretical framework on this issue, though it has been discussed extensively (at least in EE). Also, authors should address the alternatives and possible challenges regarding that. These include problems with school structrures, teachers cabililities to teach multi-desciplinary topics etc. See atleast:

Stevenson, R. B. (2007). Schooling and environmental/sustainability education: From discourses of policy and practice to
discourses of professional learning. Environmental Education Research, 13(2), 265–285. doi:10.1080/13504620701295650

Papadimitriou, V. (2004). Prospective primary teachers’ understanding of climate change, greenhouse effect, and ozone
layer depletion. Journal of Science Education and Technology, 13(2), 299–307. doi:10.1023/B:JOST.0000031268.72848.6d

Ratinen, I. (2016). Primary student teachers climate change conceptualization and implementation on inquiry-based and
communicative science teaching: A design research (Doctoral thesis). University of Jyvaskyla, Finland: Jyvaskyla Studies
in Education, Psychology and Social Research.

The authors should acknowledge that the examined VCE study designs do in fact, together address all the four "nature of CC" aspects. Ofcourse, if not all the examined study designs are implemented, then some aspects of CC may be left out, but maybe that is more of a problem of not having enough time to teach CC-related issues (I don't think it is this simple, but someone could make this case).

The reliability of the study

Researchers need to address the issue of reliability and validity of the analysis. Was inter-rater reliability measured? How was this done? etc.

Some additional suggestions
Results-section should refrain to results, rather than presenting authors' presumptions. E.g.
(line 540-541) "However this topic is chosen voluntarily, suggesting that teachers who were not trained in CC in their own education, may be less likely to teach the topic."

Omit lines 42-54, as they are irrelevant to this study.

Reviewer 2 Report

The article presents results of research on important topic - present in the media  all over the world.

I. My major concers are conected with literature selection. As the basis of theoretical framework the authors have decided to choose not scientific sources of information such as UNESCO documents and  other reports like IPCC. With full respect to UNESCO it is not dedicated to conduct science. The authors also assumed that CC is very mulitidisciplinary in its nature, similarly to the term sustainability, but presented referential model is somehow superficial. While some sources of information are neglected such as eg: 

1) Hamilton, L. C. (2011). Education, politics and opinions about climate change evidence for interaction effects. Climatic Change104(2), 231-242.

2) Pruneau, D., Gravel, H., Bourque, W., & Langis, J. (2003). Experimentation with a socio-constructivist process for climate change education. Environmental Education Research9(4), 429-446.

Additionally selected indicators are not divided into direct and indirect, there are also some indirect that are neglected by authors - like desertification and increased steppe formation etc. 

II. Methodological and results parts

1) I find the tables as very nice presented and organised, they for sure required a lot of work

2) but at the same time I am lost with the design of your studies. YYou claim that you have conducted discourse analysis - but with who? Who was the target? What tools was used for analysis etc.

3) I would be surprised if, since you don't have subject climate change, on any of these classes full requirements were covered - they re not designed for that, instead students can aquire some information on every subjects, from different perspective - that should be the case. I may not understand why you wanted any of these subjects to cover all the content? also why biology or environental science is not on the list of analysed subjects.

4) in the theoretical framework, that is quite wide, in my opinion the major point is missing, that is what it's so crucial for students to be well equiped and informed in the term of understanding the meaning of CC (as future citizens). What is also missing that climate as well as climate change it is a complex system - and as many research shows it is very difficult to atchieve the full understanding of that phenomena, even when the content is not interdyscyplinary (and in this case is additionally interdyscyplinary) - so your finding with such perspective are not surpising at all and do not add any new information to what is already known in the field. 

5) I would recommend to produce some graphical representations showing the idea of your methods used - because right now it is chaotic, it is difficult for a reader to follow your way of thinking and every steps in the research design

III. I find it very arrogant, that the authors judge others researchers results and called them anecdotal - and it is repreated 6 times in the manuscript!!!

Reviewer 3 Report

Please find my comments concerning the manuscript: “Examination of Climate Change Conceptualization within Upper Secondary Victorian Curriculum”

The idea of the paper concerning Climate Change is up to date. The article has an appropriate focus. This study affirms the readers that the secondary school curriculum in Australia does not clearly support CC education. The results can at least partly be implemented to other countries, e.g. to the reviewer´s country. How to implement a better “model” for integrating CC to school curricula was not the focus in this study, however it could have been discussed.

I can recommend this manuscript to be published with minor changes. This includes the clarification of the structure of the manuscript.

The content of the abstract is appropriate and includes normally recommended information such as the aims.

The analysis consisted of three preparatory stages aiming at identifying: study designs that address CC; characteristics of the nature of CC; and, the expected scope of CC content knowledge.

Introduction could be shortened.

In this section no aims are presented (add the aims). The aim should be presented in introduction to focus the reading of the article.

You write about the gaps “The present article seeks to address some of these knowledge gaps by focusing attention on the Australian curriculum. We are interested in developing understanding regarding potential gaps of various types. These include: gaps between the nature of climate change and its representation within the curriculum; gaps between the scope of CC as it is understood by the scientific community, and the scope of CC present in the curriculum; and, the gap between what may be regarded as ‘best practice’ curriculum integration and the current approach to integration.”

Study questions are introduced.

How is climate change conceptualised? What is the scope of CC content knowledge present in the study designs that were identified as addressing CC? How is the cross-curriculum integration approach addressed by the examined study designs?

For me it is difficult to follow the idea between aims, gaps and study questions in the manuscript.

Literature review (section 2) could be shortened Methods (Material and methods)

IPCC report was used for its reliability. What were the used methods to find the key words in the written text. (Manually, using Boolean system, or how???)

Table 1. Climate change (CC) education… Please also clarify abbreviations e.g. GHG

The explain the reliability and validity of the methods.

Results

It is easier to read and understand the results, if they have been written according to the study questions. Now you write the results according to subject, not according to the study questions, or gaps (cf. discussion). I recommend to write the results according to the study questions.

Table 2. The text in the upper part of the table (Nature of CC) is does not show at least in my version?

Table 3. The text is too small. The abbreviations CC and VCE should be explained.

Discussion

In this section you write the text according to gaps. (cf. results, aims)…Please clarify the scope of the article between aims, study questions and gaps.

I hope my comments help the authors to structure and develop their manuscript further.

Round 2

Reviewer 1 Report

The authors have made extensive changes to the article and it has improved significantly. The theoretical framework gives now a more complete picture of the existing research, also giving support for the research conducted by the authors. This has also meant that the manuscripts length has increased quite significantly and is currently too long. I believe that authors should revisit the theoretical framework and consider what are the most significant things for this particular article. Also, there is some overlap with the results section as I will discuss below in more detail. I believe the authors could reduce the length of the article by several pages, without leaving out anything relevant for the study. Reducing the length will also make the article a better read.

Reliability

line 541 indicates that the percentage of inter-rater agreement was 85%. However, percentages don’t take into consideration the randomness factor and typically are not considered a good measure for inter-rater agreement. The authors should explore using Kohen’s kappa or Krippendorfs alpha for a more informative assessment of agreement.

Results

The sections “Results part A” and “Results part B” are not really research results, and so they should not be under the Results title. In the previous version they were under the title “prepatory phase”, which is better. However, I think that the content under “Results part A” should be incorporated into the theoretical framework, as it is mainly repeating what has already been said. I see the subtitles (4.1.1, 4.1.2. etc.) as a useful way to categorize the existing research and are necessary for the “results part C”-section. Therefore, I suggest that the theoretical framework is reorganized so that these titles appear there already. This will also help avoid repetition and shorten the manuscript.

654-655 I assume these themes come from literature and are somewhat addressed in your literature review. However, as the literature review has not been conducted solely from the focus of these themes, it may be good to refer to some articles where these themes come from (reference between table and literature is made in line 760, but not in line 654-655). Also, where are the fundamental questions derived from? I think some references are needed to explain where these questions come from and why they are fundamental. It may be interesting to note that these themes and questions are similar than the themes found in high-school students’ questions on climate change (Tolppanen & Aksela, 2018).

706-707 mentions that it is a misconception that astronomical and solar systems are responsible for climate change. As there is abundant scientific evidence that these do have an affect on the change in climate, authors should explain what the problem may be in how this is presented in the Environmental Science design. Why is this said to be a misconception?  If the authors do not want to expand the article further, this mention of misconceptions could also be omitted to avoid confusion.

707-708 The second misconception also calls for further explanation.

Presentation of findings in much clearer now.

Discussion &Conclusions

line 885-886 The authors claim that “Our evidence clearly indicates that this approach (cross-curriculum) is ineffective”. I disagree with this claim and the conclusions that follow. The data of this study merely shows that holistic CC-education has not been holistically implemented into their curriculum. As the implementation has failed, it is too early to say whether cross-curriculum implementation if ineffective.

In my view, this claim can only be made IF the curriculum were well balanced on CC-issues BUT students would still fail to get a holistic picture of CC. However, that is beyond the scope of this study.

As the authors have noted, CC is a multifaceted issue, including scientific, societal and ethical issues. They also note that teachers lack expertise in teaching on CC-issues (line 895). Yet they “propose establishing a new curriculum subject dedicated to CC education” (898), without considering the challenges of this. In practice, this would mean that the teacher of such a subject would need to be an expert on scientific, societal and ethical issues, would be able to address values, help develop thinking skills, support participatory action and deal with students eco-anxiety, to name a few of the needed competencies (inline with Tolppanen et al,. 2017; Cantell et al, 2019). It is not certain at all that a single teacher could deal with all of these dimensions of CC-education, which could result in integrated CC-education not being holistic after all. Also, training such teachers would put a big strain on teacher education. Where would they get their training from? Who is competent to teach them? That said, simply changing the structure of CC from cross-curricular to subject-based education will not solve the problems of CC-education and the authors should acknowledge this in their discussion and conclusions.

As a general rule, the discussion section should not present new information, but rather discuss the findings in relation to previous research. However, in lines 938- 980 the authors go about presenting some of the questions and statements in the packages they have examined. If the authors wish to include quotes from the different work-packages etc., I think these would fit better into the results section. However, they would also need to be analyzed systematically and therefore, I don’t think they fit into the scope of this study. I also find much of the discussion on these questions to be somewhat naïve or unjustified, as I will explain through the few examples below:

938-948 The authors claim that the question presents “fundamental lack of understanding”. However, my question is, how can this question (or any question for that matter) show a lack of understanding? I don’t agree with the authors analysis that the question implies anything. Sure, the question is simple to answer and a more in-depth answer could be obtained by asking why the barriers cannot be quarantined (or how could they), rather than merely asking can they, because the latter results in simplistic yes/no answers. However, though the question may be poorly written, I think the authors go too far when they claim that this shows a lack of understanding of the science behind the issue. To me it only shows that the writer of the question does not know how to formulate questions that start a meaningful discussion/inquiry.

951-955 Another possible misjudgement of the intent of a question is shown in lines 951-955. Again, the question itself does not necessarily try to imply anything, contrary to what the authors suggest. Maybe this is merely a question aimed to get classroom discussion going? I don’t see what is wrong in this, though again, the question could be framed differently to produce more insight into discussion, rather than becoming a Yes vs. No debate.

I don’t find lines 938-980 to be a meaningful discussion of the findings, and therefore, I think it would be better to omit these.

Minor notes

151-154 specifies the type of curricula of certain countries. However, the reference is almost 15 years old, meaning that his information is outdated. For this reason, I think you should leave out specifying any countries, as curricula have certainly changed. Alternatively, you can look for more resent publications on the curricula of the mentioned countries.

line 188 Should this read “not encouraging” rather than “encouraging”

line 196 “the” is written two times

line 333 (+810 etc.) The original source of the bicycle model is Tolppanen et al., 2017, as acknowledged in Cantell et al., 2019

Reviewer 2 Report

The manuscript has been improved in many ways, and its clarity is much better now. I am glad about that, especially that this article should be read not only by environmental education researchers as the authors claim, but it should be read by policy makers as well as teachers and students. In such way presented article is part of the growing field of evidence-based educational research in environmental education.

In their response the authors summarised their novelty in 4 points

For the first time to the best of our knowledge the nature of CC is conceptualised We developed CC scoping map. Thus far all the educational models that we have seen did not address the contents as separate from other educational goals such as students’ empowerment. We analysed for the first time to our knowledge the upper secondary curriculum in regard to implementing CC. We provide evidence that the cross-curriculum approach doesn’t work.  All of them are in the text, but in my opinion can be more explicit. Especially that some of the findings are quite surprising and dramatic/sad. so the article can be accepted in present form, but I would recommend to add one strong sentence in abstract and in conclusions. 
